# Wasserstein Dependency Measure
# for Representation Learning

**Sherjil Ozair**
Mila, Université de Montréal

**Corey Lynch**
Google Brain

**Yoshua Bengio**
Mila, Université de Montréal

**Aäron van den Oord**
Deepmind

**Sergey Levine**
Google Brain

**Pierre Sermanet**
Google Brain

## Abstract

Mutual information maximization has emerged as a powerful learning objective for unsupervised representation learning obtaining state-of-the-art performance in applications such as object recognition, speech recognition, and reinforcement learning. However, such approaches are fundamentally limited since a tight lower bound on mutual information requires sample size exponential in the mutual information. This limits the applicability of these approaches for prediction tasks with high mutual information, such as in video understanding or reinforcement learning. In these settings, such techniques are prone to overfit, both in theory and in practice, and capture only a few of the relevant factors of variation. This leads to incomplete representations that are not optimal for downstream tasks. In this work, we empirically demonstrate that mutual information-based representation learning approaches do fail to learn complete representations on a number of designed and real-world tasks. To mitigate these problems we introduce the Wasserstein dependency measure, which learns more complete representations by using the Wasserstein distance instead of the KL divergence in the mutual information estimator. We show that a practical approximation to this theoretically motivated solution, constructed using Lipschitz constraint techniques from the GAN literature, achieves substantially improved results on tasks where incomplete representations are a major challenge.

## 1 Introduction

Recent success in supervised learning can arguably be attributed to the paradigm shift from engineering representations to learning representations [32]. Especially in the supervised setting, effective representations can be acquired directly from the labels. However, representation learning in the unsupervised setting, without hand-specified labels, becomes significantly more challenging: although much more data is available for learning, this data lacks the clear learning signal that would be provided by human-specified semantic labels.

Nevertheless, unsupervised representation learning has made significant progress recently, due to a number of different approaches. Representations can be learned via implicit generative methods [22, 17, 16, 40], via explicit generative models [28, 44, 13, 44, 29], and self-supervised learning [6, 15, 50, 14, 47, 49, 24]. Among these, the latter methods are particularly appealing because they remove the need to actually generate full observations (e.g., image pixels or audio waveform). Self-supervised learning techniques have demonstrated state-of-the-art performance in speech and image understanding [47, 24], reinforcement learning [25, 18, 26], imitation learning [45, 5], and natural language processing [12, 43].

Self-supervised learning techniques make use of discriminative *pretext* tasks, chosen in such a way that its labels can be extracted automatically and such that solving the task requires a semantic understanding of the data, and therefore a meaningful representation. For instance, Doersch et al. [15] predict the relative position of adjacent patches extracted from an image. Zhang et al. [50] reconstruct images from their grayscaled versions. Gidaris et al. [21] predict the canonically upwards direction in rotated images. Sermanet et al. [45] maximize the mutual information between two views of the same scene. Hjelm et al. [24] maximize mutual information between an image and patches of the image.

However, a major issue with such techniques is that the pretext task must not admit trivial or easy solutions. For instance, Doersch et al. [15] found that relative position could be easily predicted using low-level cues such as boundary patterns, shared textures, long edges, and even chromatic aberration. In some cases, trivial solutions are easily identifiable and rectified, such as by adding gaps between patches, adding random jitter, and/or grayscaling the patches.

Identifying such exploits and finding fixes is a cumbersome process, requires expert knowledge about the domain, and can still fail to eliminate all degenerate solutions. However, even when care is taken to remove such low-level regularities, self-supervised representation learning techniques can still suffer and produce incomplete representations, i.e. representations that capture only a few of the underlying factors of variations in the data.

Recent work has provided a theoretical underpinning of this empirically observed shortcoming. A number of self-supervised learning techniques can be shown to be maximizing a lower bound to mutual information between representations of different data modalities [7, 47, 42]. However, as shown by McAllester and Statos [35], lower bounds to the mutual information are only tight for sample size exponential in the mutual information. Unfortunately, many practical problems of interest where representation learning would be beneficial have large mutual information. For instance, mutual information between successive frames in a temporal setting scales with the number of objects in the scene. Self-supervised learning techniques in such settings often only capture a few objects since modeling a few objects is sufficient to confidently predict future frames from a random sample of frames.

In this paper, we motivate this limitation formally in terms of the fundamental limitations of mutual information estimation and KL divergences, and show examples of this limitation empirically, illustrating relatively simple problems where fully reconstructive models can easily learn complete representations, while self-supervised learning methods struggle. We then propose a potential solution to this problem, by employing the Wasserstein metric in place of KL divergence as a training objective. In practice, we show that approximating this by means of recently proposed regularization methods designed for generative adversarial networks can substantially reduce the incomplete representation problem, leading to a substantial improvement in the ability of representations learned via mutual information estimation to capture task-salient features.

## 2  Mutual Information Estimation and Maximization

Mutual information for representation learning has a long history. One approach [33, 8] is to maximize the mutual information between observed data samples $x$ and learned representations $z = f(x)$, i.e. $I(x; z)$, thus ensuring the representation learned retain the most information about the underlying data. Another [6] is to maximize the mutual information between representations of two different modalities of the data, i.e. $I(f(x); f(y))$.

However, such representation learning approaches have been limited due to the difficulty of estimating mutual information. Previous approaches have had to make parametric assumptions about the data or use nonparametric approaches [30, 37] which don't scale well to high-dimensional data.

More recently Nguyen et al. [39], Belghazi et al. [7], van den Oord et al. [47], and Poole et al. [42] have proposed variational energy-based lower bounds to the mutual information which are tractable in high dimension and can be estimated by gradient-based optimization, which makes them suitable to combine with deep learning.

While Becker and Hinton [6] maximize mutual information between representations of spatially adjacent patches of an image, one can also use past and future states such as shown recently by van den Oord et al. [47] and Sermanet et al. [45] which has connections to predictive coding in speech [4, 19], predictive processing in the brain [10, 41, 46] and the free energy principle [20].

These techniques have shown promising results, but their applicability is still limited to low mutual information settings.

## 2.1 Formal Limitations in Mutual Information Estimation

The limitations of estimating mutual information via lower bounds stems from the limitations of the KL divergence as a measure of distribution similarity. Theorem 1 formalizes this limitation of estimating the KL divergence via lower bounds. This result is based on the derivation by McAllester and Statos [35], who prove a stronger claim for the case where $p(x)$ is fully known.

**Theorem 1.** *Let $p(x)$ and $q(x)$ be two distributions, and $R = \{x_i \sim p(x)\}_{i=1}^n$ and $S = \{x_i \sim q(x)\}_{i=1}^n$ be two sets of $n$ samples from $p(x)$ and $q(x)$ respectively. Let $\delta$ be a confidence parameter, and let $B(R, S, \delta)$ be a real-valued function of the two samples $S$ and $R$ and the confidence parameter $\delta$. We have that, if with probability at least $1 - \delta$,*

$$B(R, S, \delta) \leq KL(p(x)\|q(x))$$

*then with probability at least $1 - 4\delta$ we have*

$$B(R, S, \delta) \leq \log n.$$

Thus, since the mutual information corresponds to KL divergence, we can conclude that any high-confidence lower bound on the mutual information requires $n = \exp(I(x; y))$, i.e., sample size exponential in the mutual information.

# 3 Wasserstein Dependency Measure

The KL divergence is not only problematic for representation learning due to the statistical limitations described in Theorem 1, but also due to the fact that it is completely agnostic to the metric of the underlying data distribution, and invariant to any invertible transformation. KL divergence is also sensitive to small differences in the data samples. When used for representation learning, the encoder can often only represent small parts of the data samples, since any small differences found is sufficient to maximize the KL divergence. The Wasserstein distance, however, is a metric-aware divergence, and represents the difference between two distributions in terms of the actual distance between data samples. A large Wasserstein distance actually represents large distances between the underlying data samples. On the other hand, KL divergence can be large even if the underlying data samples differ very little.

This qualitative difference between the KL divergence and Wasserstein distance was recently noted by Arjovsky et al. [3] to propose the Wasserstein GAN, a metric-aware extension to the original GAN proposed by Goodfellow et al. [22] which is based on the Jensen symmetrization of the KL divergence [11]. For GANs, we would like the discriminator to model not only the density ratio of two distributions, but the complete process of how one distribution can be transformed into another, which is the underlying basis of the theory of optimal transport and Wasserstein distances [48].

This motivates us to investigate the use of the Wasserstein distance as a replacement for the KL divergence in mutual information, which we call *Wasserstein dependency measure*.

**Definition 3.1. Wasserstein dependency measure**. For two random variables $x$ and $y$ with joint distribution $p(x, y)$, we define the Wasserstein dependency measure $I_{\mathcal{W}}(x; y)$ as the Wasserstein distance between the joint distribution $p(x, y)$ and the product of marginal distributions $p(x)p(y)$.

$$I_{\mathcal{W}}(x; y) \stackrel{\text{def}}{=} \mathcal{W}(p(x, y), p(x)p(y)) \tag{1}$$

Thus, the Wasserstein dependency measure (WDM) measures the cost of transforming samples from the marginals to samples from the joint, and thus cannot ignore parts of the input, unlike the KL divergence which can ignore large parts of the input.

**Choice of Metric Space.** The Wasserstein dependency measure assumes that the data lies in a known metric space. However, the purpose of representation learning is, in a sense, to use the representations to implicitly form a metric space for the data. Thus, it may seem that we're assuming the solution by requiring knowledge of the metric space. However, the difference between the two is

that the base metric used in the Wasserstein distance is data-independent, while the metric induced by the representations is informed by the data. The two metrics can be thought of as prior and posterior metrics. Thus, the base metric should encode our prior beliefs about the task independent of the data samples, which acts as inductive bias to help learn a better posterior metric induced by the learned representations. In our experiments we assume a Euclidean metric space for all the tasks.

## 3.1 Generalization of Wasserstein Distances

Theorem 1 is a statement about the inabiltiy of mutual information bounds to generalize for large values of mutual information, since the gap between the lower bound sample estimate and the true mutual information can be exponential in the mutual information value. The Wasserstein distance, however, can be shown to have better generalization properties when used with Lipschitz neural net function approximation via its dual representation. Neyshabur et al. [38] show that a neural network's generalization gap is proportional to the square root of the network's Lipschitz constant, which is bounded $(= 1)$ for the function class used in Wasserstein distance estimation, but is unbounded for the function class used in mutual information lower bounds.

## 4 Wasserstein Predictive Coding for Representation learning

Estimating Wasserstein distances is intractable in general. We will use the Kantorovich-Rubenstein duality [48] to obtain the dual form of the Wasserstein dependency measure, which allows for easier estimation since the dual form allows gradient-based optimization over the function space using neural networks.

$$
\begin{aligned}
I_{\mathcal{W}}(x; y) &\overset{\text{def}}{=} \mathcal{W}(p(x, y), p(x)p(y)) \\
&= \sup_{f \in \mathcal{L}_{\mathcal{M} \times \mathcal{M}}} \mathbb{E}_{p(x,y)}[f(x, y)] - \mathbb{E}_{p(x)p(y)}[f(x, y)]
\end{aligned}
\tag{2}
$$

Here, $\mathcal{L}_{\mathcal{M} \times \mathcal{M}}$ is the set of all 1-Lipschitz functions in $\mathcal{M} \times \mathcal{M} \to \mathbb{R}$. We note that Equation 2 is similar to contrastive predictive coding (CPC) [47], which optimizes

$$
\begin{aligned}
\mathcal{J}_{CPC} &= \sup_{f \in \mathcal{F}} \mathbb{E}_{p(x,y)p(y_j)} \left[ \log \frac{\exp f(x, y)}{\sum_j \exp f(x, y_j)} \right] \\
&= \sup_{f \in \mathcal{F}} \mathbb{E}_{p(x,y)}[f(x, y)] - \mathbb{E}_{p(x)p(y_j)} \left[ \log \sum_j \exp f(x, y_j) \right].
\end{aligned}
\tag{3}
$$

The two main differences between contrastive predictive coding and the dual Wasserstein dependency measure is the Lipschitz constraint on the function class and the $\log \sum \exp$ in the second term of CPC.

We propose a new objective, which is a lower bound on both contrastive predictive coding and the dual Wasserstein dependency measure, by keeping both the Lipschitz class of functions and the $\log \sum \exp$, which we call Wasserstein predictive coding (WPC):

$$
\mathcal{J}_{WPC} = \sup_{f \in \mathcal{L}_{\mathcal{M} \times \mathcal{M}}} \mathbb{E}_{p(x,y)}[f(x, y)] - \mathbb{E}_{p(x)p(y_j)} \left[ \log \sum_j \exp f(x, y_j) \right].
\tag{4}
$$

We choose to keep the $\log \sum \exp$ term, since it decreases the variance when we use samples to estimate the gradient, which we found to improve performance in practice. In the previous sections, we motivated the use of Wasserstein distance, which directly suggests the use of a Lipschitz constraint in Equation 4. CPC and similar contrastive learning techniques work by reducing the distance between paired samples, and increasing the distance between unpaired random samples. However, when using powerful neural networks as the representation encoder, the neural network can learn to exaggerate small differences between unpaired samples to increase the distance between arbitrarily. This then prevents the encoder to learn any other differences between unpaired samples because one discernible difference suffices to optimize the objective. However, if we force the encoder to be Lipschitz, then the distance between learned representations is bounded by the distance between the underlying samples. Thus the encoder is forced to represent more components of the data to make progress in the objective.

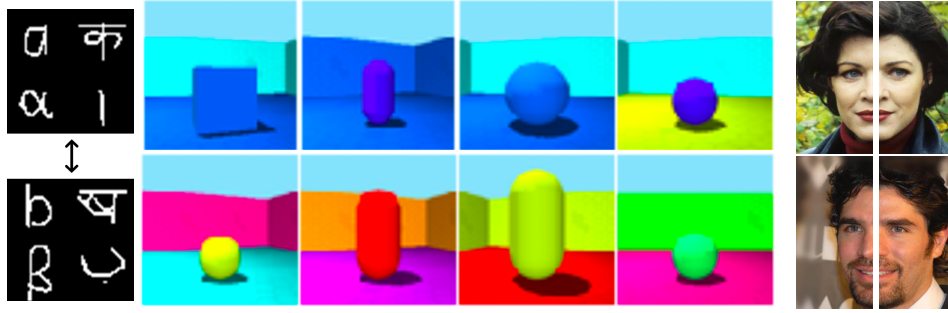

Figure 1: **Left** - The SpatialMultiOmniglot dataset consists of pairs of images $(x, y)$ each comprising of multiple Omniglot characters in a grid, where the characters in $y$ are the next characters in the alphabet of the characters in $x$. **Middle** - The Shapes3D dataset is a collection of colored images of an object in a room. Each image corresponds to a unique value for the underlying latent variables: color of object, color of wall, color of floor, shape of object, size of object, viewing angle. **Right** - The SplitCelebA dataset consists of pairs of images $p(x, y)$ where $x$ and $y$ are the left and right halves of the same CelebA image, respectively.

## 4.1 Approximating Lipschitz Continuity

Optimization over Lipschitz functions with neural networks is a challenging problem, and a topic of active research. Due to the popularity of the Wasserstein GAN [3], a number of techniques have been proposed to approximate Lipschitz continuity [23, 36]. However, recent work [2] has shown that that such techniques severely restrict the capacity of typical neural networks and could hurt performance in complex tasks where high capacity neural networks are essential. This is also observed empirically by Brock et al. [9] in the context of training GANs.

Thus, in our experiments, we use the gradient penalty technique proposed by Gulrajani et al. [23], which is sufficient to provide experimental evidence in support of our hypothesis, but we note the caveat that gradient penalty might not be effective for complex tasks. Incorporating better and more scalable methods to enforce Lipschitz continuity would likely further improve practical WDM implementations.

## 5 Experiments

The goal of our experiments are the following. We demonstrate and quantify the limitations of mutual information-based representation learning. We quantitatively compare our proposed alternative, the Wasserstein dependency measure, with mutual information for representation learning. We demonstrate the importance of the class of functions being used to practically approximate dependency measures, such as fully-connected or convolutional networks.

### 5.1 Evaluation Methodology

All of our experiments make use of datasets generated via the following process: $p(z)p(x|z)p(y|z)$. Here, $z$ is the underlying latent variable, and $x$ and $y$ are observed variables. For example, $z$ could be a class label, and $x$ and $y$ two images of the same class. We specifically use datasets with large values of the mutual information $I(x; y)$, which is common in practice and is also the condition under which we expect current MI estimators to struggle.

The goal of the representation learning task is to learn representation encoders $f \in \mathcal{F}$ and $g \in \mathcal{F}$, such that the representations $f(x)$ and $g(y)$ capture the underlying generative factors of variation represented by the latent variable $z$. For example, for SpatialMultiOmniglot (described in 5.2), we aim to learn $f(x)$ which captures the class of each of the characters in the image. However, representation learning is not about making sure the representations contain the requisite information, but that they contain the requisite information in an accessible way, ideally via linear probes [1]. Thus, we measure the quality of the representations by learning linear classifiers predicting the

Table 1: WPC outperforms CPC on the SplitCelebA dataset.

| Method | CPC (fc) | WPC (fc) | CPC (conv) | WPC (convnet) |
|--------|----------|----------|------------|---------------|
| Accuracy | 0.85 | **0.87** | 0.82 | **0.87** |

underlying latent variables $z$. This methodology is standard in the self-supervised representation learning literature.

## 5.2 Datasets

We present experimental results on four tasks, SpatialMultiOmniglot, StackedMultiOmniglot, MultiviewShapes3D, and SplitCelebA.

**SpatialMultiOmniglot.** We used the Omniglot dataset [31] as a base dataset to construct Spatial-MultiOmniglot and StackedMultiOmniglot. SpatialMultiOmniglot is a dataset of paired images $x$ and $y$, where $x$ is an image of size $(32m, 32n)$ comprised of $mn$ Omniglot character arranged in a $(m, n)$ grid from different Omniglot alphabets, as illustrated in Figure 1. The characters in $y$ are the next characters of the corresponding characters in $x$, and the latent variable $z$ is the index of each of the characters in $x$.

Let $l_i$ be the alphabet size for the $i^{\text{th}}$ character in $x$. Then, the mutual information $I(x; y)$ is $\sum_{i=1}^{mn} \log l_i$. Thus, adding more characters increases the mutual information and allows easy control of the complexity of the task.

For our experiments, we picked the 9 largest alphabets which are Tifinagh (55 characters), Japanese (hiragana) (52 characters), Gujarati (48), Japanese (katakana) (47), Bengali (46), Grantha (43), Sanskrit (42), Armenian (41), and Mkhedruli (Georgian) (41), with their respective alphabet sizes in parentheses.

**StackedMultiOmniglot.** StackedMultiOmniglot is similar to SpatialMultiOmniglot except the characters are stacked in the channel axis, and thus $x$ and $y$ are arrays of size $(32, 32, n)$. This dataset is designed to remove feature transfer between characters of different alphabets that is present in SpatialMultiOmniglot when using convolutional neural networks. The mutual information is the same, i.e. $I(x; y) \sum_{i=1}^{n} \log l_i$.

**MultiviewShapes3D.** Shapes3D [27] (Figure 1) is a dataset of images of a single object in a room. It has six factors of variation: object color, wall color, floor color, object shape, object size, and camera angle. These factors have 10, 10, 10, 4, 6, and 15 values respectively. Thus, the total entropy of the dataset is $\log(10 \times 10 \times 10 \times 4 \times 6 \times 15)$. MultiviewShapes3D is a subset of Shapes3D where we select only the two extreme camera angles for $x$ and $y$, and the other 5 factors of $x$ and $y$ comprise the latent variable $z$.

**SplitCelebA.** CelebA [34] is a dataset consisting of celebrity faces. The SplitCelebA task uses samples from this dataset split into left and right halves. Thus $x$ is the left half, and $y$ is the right half. We use the CelebA binary attributes as the latent variable $z$.

## 5.3 Experimental Results

**Effect of Mutual Information.** Our first main experimental contribution is to show the effect of dataset size on the performance of mutual information-based representation learning, in particular, of contrastive predictice coding (CPC). Figure 2 (bottom) shows the performance of CPC and WPC as the mutual information increases. We were able to control the mutual information in the data by controlling the number of characters in the images. We kept the training dataset size fixed at 50,000 samples. This confirms our hypothesis that mutual information-based representation learning indeed suffers when the mutual information is large. As can be seen, for small number (1 and 2) of characters, CPC has near-perfect representation learning. The exponential of the mutual information in this case is 55 and $55 \times 52 = 2860$ (i.e. the product of alphabet class sizes), which is smaller than the dataset size. However, when the number of characters is 3, the exponential of the mutual information is $55 \times 52 \times 48 = 137280$ which is larger than the dataset size. This is the case where

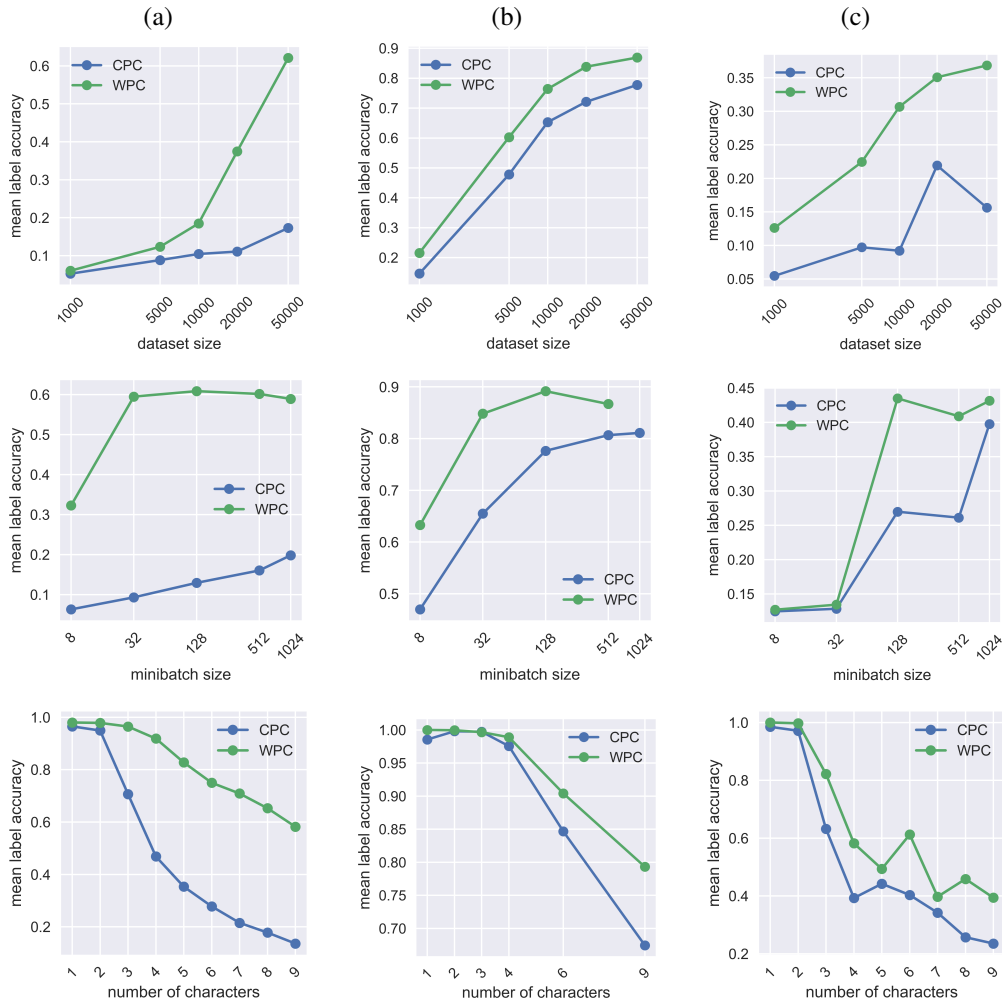

Figure 2: Performance of CPC and WPC **(a)** on SpatialMultiOmniglot using fully-connected neural networks, **(b)** on StackedMultiOmniglot using fully-connected networks, and **(c)** using convolutional neural networks. **Top** - WPC consistently performs better than CPC over different dataset sizes, especially when using fully-connected networks. **Middle** - WPC is more robust to minibatch size, while CPC's performance drops rapidly on reduction in minibatch size. **Bottom** - As mutual information is increased, WPC's drop in performance is more gradual, while CPC's drop in performance is drastic (when mutual information passes log dataset size).

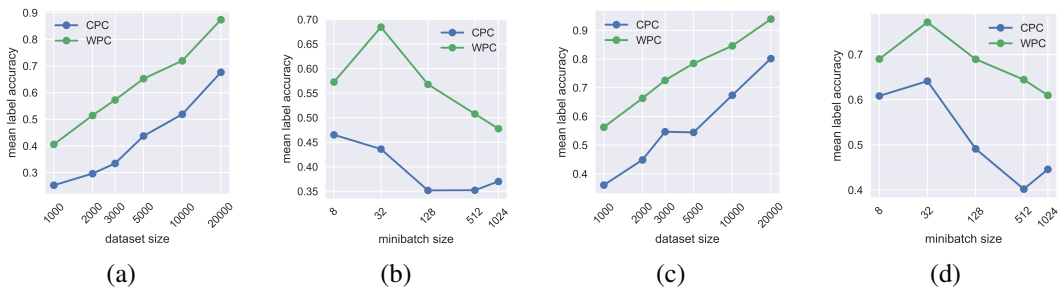

Figure 3: Performance of CPC and WPC on MultiviewShapes3D using (a,b) fully-connected networks, and (c,d) convolutional network. WPC performs consistently better than CPC for multiple dataset and minibatch sizes.

CPC is no longer a good lower bound estimator for the mutual information, and the representation learning performance drops down significantly.

We observe that while WPC's performance also drops when the mutual information is increased, however it's always better than CPC, and the drop in performance is more gradual. Ideally, representation learning performance should not be effected by the number of characters at all. We believe WPC's less-than-ideal performance is due to the practical approximations we used such as gradient penalty.

**Effect of Dataset Size.** Figures 2 (top), and 3 (a,c) show the performance of CPC and WPC as we vary the dataset size. For the Omniglot datasets, the number of characters has been fixed to 9, and the mutual information for this dataset is the logarithm of the product of the 9 alphabet sizes which is around 34.43 nats. This is a very large information value as compared to the dataset size, and thus, we observe that the performance of either method is far from perfect. However, WPC performs significantly better than CPC.

**Effect of Minibatch Size.** Both CPC and WPC are minibatch-dependent techniques. For small mininatches, the variance of the estimator becomes large. We observe in Figure 2 (middle) that CPC's performance increases as the minibatch size is increased. However, WPC's performance is not as sensitive to the minibatch size. WPC reaches its optimal performance with a minibatch size of 32, and any further increase in minibatch size does not improve the performance. This suggests that Wasserstein-based representation learning can be effective even at small minibatch sizes.

**Effect of Neural Network Inductive Bias.** The use of fully connected neural networks allowed us to make predictions about the performance based on whether the mutual information is larger or smaller than the log dataset size. However, most practical uses of representation learning use convolutional neural networks (convnet). Convnets change the interplay of mutual information and dataset sizes, since they can be more efficient with smaller dataset sizes since they bring in their inductive biases such as translation invariance or invertibility via residual connections. Convnets also perform worse on StackedMultiOmniglot (Figure 2 (c)) than on SpatialMultiOmniglot (Figure 2 (b)), which is expected since SpatialMultiOmniglot arranges the Omniglot characters spatially which works well with convnet's translation invariance. When the data does not match convnet's inductive bias, as in StackedOmniglot, WPC provides a larger improvement over CPC.

# 6 Conclusion

We proposed a new representation learning objective as an alternative to mutual information. This objective which we refer to as the Wasserstein dependency measure, uses the Wasserstein distance in place of KL divergence in mutual information. A practical implementations of this approach, Wasserstein predictive coding, is obtained by regularizing existing mutual information estimators to enforce Lipschitz continuity. We explore the fundamental limitations of prior mutual information-based estimators, present several problem settings where these limitations manifest themselves, resulting in poor representation learning performance, and show that WPC mitigates these issues to a large extent. However, optimization of Lipschitz-continuous neural networks is still a challenging problem. Our results indicate that Lipschitz continuity is highly beneficial for representation learning, and an exciting direction for future work is to develop better techniques for enforcing Lipschitz continuity. As better regularization methods are developed, we expect the quality of representations learned via Wasserstein dependency measure to also improve.

# Acknowledgement

The authors would like to thank Ben Poole, George Tucker, Alex Alemi, Alex Lamb, Aravind Srinivas, Ethan Holly, Eric Jang, Luke Metz, and Julian Ibarz for useful discussions and feedback on our research. SO is thankful to the Google Brain team for providing a productive and empowering research environment.

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
