[Reviews · NeurIPS 2019]

Reviewer 1



In this paper the authors discuss the limitations of current representation learning approaches that aim to approximate MI, and propose their own objective, a lower bound on what they call the Wasserstein dependency measure. They compare their method to a similar approach, CPC, where the difference largely lies in the constraint that the function class they are optimizing over is 1-Lipschitz. Their results are really promising and show increased robustness and performance on small, relatively simple datasets with regards to dataset size, mini batch size and mutual information of the dataset. Overall, I think this is a nice paper - it's a simple modification to an existing approach, but the performance gains are significant in the settings considered. Id like to see more discussion on the limitations of this method in more complicated settings - this is alluded in the paper, but I think careful analysis about the limitations of an approach is also valuable in addition to the analysis about its strengths.

Reviewer 2



I like the idea of wassersteinizing Mutual information, and the reasoning that enforcing Lipschitz constraint in CPC bounds prevents the encoder to exaggerate small differences. Although, I am not fully convinced if the underlying problem with exponential sample size is resolved by their lower bound. Besides, there are some questions on clarity and experiments which I believe the authors need to address. >>>>> Questions on experiments: - What would happen if you directly used the proposed Wasserstein dependency measure instead of considering any contrastive learning/negative samples? You won't have la ower bound, and this proposed Wasserstein mutual information would be exact. - "illustrating relatively simple problems where fully reconstructive models can easily learn complete representations, while self-supervised learning methods struggle." In which experiments, do you show that fully reconstructive models can easily learn...? - In the CPC paper, they report top1 and top5 unsupervised classification results on ImageNet. Are there any reasons that prevent experiments with WPC in this setting? >>>>> Quality and Clarity: - How is the base metric (which is typically called ground metric in OT literature) in Wasserstein data independent? For instance, if the distribution is over words, it makes more sense to use angular/cosine distance between word embeddings. I agree with what you say in lines 118-120, and maybe it is just the particular wording of the line 116 that needs to be refined. - From my understanding, the summation over j in eq (3) is over the set (say Y) containing the target y and the other negatives samples. So in the second term of eq(3), the expectation is over some p(x)p(y_j) for some j, while inside the sum is over all j indexing the set Y. Maybe, I missing something but doesn't seem you can take out the expectation over some j like that. (I understand what you want to do from this, but I think the equation is not precise). - Also, it would be useful to provide some background on CPC in your paper. It will make it more complete. - Besides the two main differences mentioned, aren't equation (2) and (3) also different in the sense that (2) doesn't have second term (expectation of log sum exp) for negative samples? If yes, I think it would be good to clarify in the text. - How is the reference section organized? It is neither ordered by serial numbers, nor ordered by last names or first names, or nor by publication date. It is really hard to check! >>>>> Typos: Line 209, I(x; y) = Some of the inline references are not properly typeset, example: "ideally via linear probes Alain and Bengio [2016]." Line 222 "predictice" -> "predictive" Line 244 "mininatches" -> "minibatches" (Some nitpicking) Line 135: \mathcal{M} is not defined. ----- UPDATE ----- Thanks for the rebuttal and it addresses most of the concerns I had. As mentioned in the rebuttal, I think it is critical to clarify that your method's efficacy is shown empirically and not theoretically. Please make the suggested revisions in the final version and *at all costs* improve the reference section.

Reviewer 3



Originality: I'm not very familiar but experimental tasks looked new to me. Moreover, the proposed approach is a novel combination of previous works. Quality: All arguments in the paper is well supported verbally. In the experiments section, they also supported their claims empirically. Clarity: Overall it is good. I had difficulties to follow 'Choice of Metric Space' subsection. Significance: I think other researchers won't use this approach as is for solving problems but they will try to build upon. I think this work can lead to other high quality publications. For methodology, it follows a similar approach borrows different components from previous related work and build upon. Although research direction looks very promising, experimental results looked very dataset/experimental setup specific to me.

[Author Response · NeurIPS 2019]

We thank the reviewers for the detailed and positive feedback on our work. We address the specification clarifications
requested by the reviewers below, with an explanation of how each of these will be addressed in the final paper. We
believe that these clarifications will resolve all of the reviewer concerns, but would be happy to take any additional
suggestions into account.

**General response regarding experiments (R3 & R4)**: Our experiments are designed specifically to study the problem
with mutual information estimation that we discuss in Sec 2.1 (see also discussion in Sec 5.3). Datasets with high
mutual information make it difficult to learn complete representations with KL-divergence-based methods like CPC.
Not all tasks have this issue, and we expect that tasks where this is not an issue would have similar performance for
CPC and WPC. We will clarify this, explaining that we are not proposing a general improvement across all tasks, but a
specific improvement that aims to address the issue that we motivate in Sec 2.1 and confirm experimentally in Fig 2
(bottom row).

**R4**: *Q: Difference in comparison of experimental setup vs. CPC? A:* As we state above, our experiments are chosen to
study settings with high mutual information, as motivated in Sec 2.1. We illustrate that this problem happens in practice
(see Fig 2, bottom row), and then show that WPC mitigates the problem. We had to design new tasks to allow us to vary
the mutual information in the data. Because of this, our experimental setup is different from the CPC paper – we are not
trying to show that WDM improves over CPC in all cases, but that it mitigates the particular problem motivated in the
paper. We will clarify this in the final version.

**R4**: *Q: code? A:* We were unable to prepare the code for release in time for submission, but will release it soon.

**R4**: *Q: Evaluation is specialized on images. A:* We agree that the method is general, and not specific to images.
However, it is very common for representation learning work to be evaluated on images. Images provide a number of
challenges and considerable breadth, and there are many standard datasets, making the visual domain well-suited for
comparing representation learning methods.

**R4**: *Q: Abstract's "real-world tasks"? A:* We will revise the abstract to replace "real-world tasks" with "a number of
tasks with synthetic and realistic images." We would be happy to make other revisions that the reviewers might suggest.

**R3**: *Q: How is the problem of exponential sample size resolved? A:* We will clarify the discussion regarding exponential
sample size. We do not have a formal result showing that the WDM actually resolves the exponential sample size
issue fully. The exponential sample size discussion is meant to motivate seeking an alternative to the KL-divergence
for representation learning, but the utility of our proposed WDM solution is verified empirically, with experiments
conducted on a set of tasks that are intentionally selected to have high mutual information between the context and
predicted variable. The experimental results show that WDM produces better performance under these conditions. We
will clarify that the primary evidence for the efficacy of our method is empirical, rather than theoretical.

**R3**: *Q: top5 and top1 results on ImageNet? A:* Unfortunately, we did not have time to conduct this experiment during
the rebuttal, though we will try to add it in the final. However, as we discuss above, the goal of our experiments is to
evaluate settings with high mutual information in a controlled way, rather than show improvement over CPC in all cases.

**R3**: *Q: Directly estimating the WDM without a bound and without contrastive/negative samples? A:* We are not aware
of any way to estimate Wasserstein distance exactly without employing a bound or a variational approximation for high
dimensional inputs, though this would certainly be an improvement to the method. We will discuss this in the paper.

**R3**: *Q: Base metric, discussion of CPC? A:* We will revise the wording on line 116 to address your suggestion. We will
also add a more complete summary of CPC to ensure that the discussion in the paper stands on its own.

**R3**: *Q: Summation over $j$ in Equation 3. A:* This notation can definitely be improved. The outer $j$ (in the expectation)
and the inner one (in the log-sum-exp) are different, but running over the same set. The inner one is the one observed in
the data, the outer one goes over all possible values. We will change the notation to clarify.

**R3**: *Q: Equations 2 and 3. A:* Both equations have negative samples (from the product of marginals), albeit their scores
are accounted in a different way, either as a simple average (Eqn 2) or as a log-sum-exp. Will clarify in the text.

**R2**: *Q: Limitations, and when does WPC break down as compared to CPC? A:* In general, we did not observe settings
where WPC "breaks down" as compared to CPC. This is likely because the difference in the actual methods is not very
large: while the theoretical basis is quite different, implementation-wise WPC amounts to CPC augmented with an
additional regularizer. The regularizer is meant to ensure that the function class is 1-Lipschitz. A variety of regularizers
can be used to get this effect, though we use gradient penalty (GP). In principle, GP might also lead to underfitting if
the regularizer has a large coefficient and the function class is too small, as with any other regularizer, but we did not
observe this problem experimentally. We will endeavor to conduct additional experiments on larger datasets for the final
version to identify other potential corner cases or limitations, and discuss the tradeoffs of introducing regularization in
the discussion section.

[Meta-Review · NeurIPS 2019]

The rebuttal addresses most the reviewers concerns and they decided that this paper should be accepted as a poster at NeurIPS.